# Overexpression of REST Represses the Epithelial–Mesenchymal Transition Process and Decreases the Aggressiveness of Prostate Cancer Cells

**DOI:** 10.3390/ijms25063332

**Published:** 2024-03-15

**Authors:** Sebastián Indo, Octavio Orellana-Serradell, María José Torres, Enrique A. Castellón, Héctor R. Contreras

**Affiliations:** 1Laboratory of Cellular and Molecular Oncology, Department of Basic and Clinical Oncology, Faculty of Medicine, University of Chile, Santiago 8380453, Chile; srindo@uchile.cl (S.I.); mjtorres@uc.cl (M.J.T.); 2Center for Cancer Prevention and Control (CECAN), Santiago 8380453, Chile; 3Millennium Nucleus of Ion Channel-Associated Diseases (MiNICAD), Santiago 8380453, Chile; octavio.orellana.s@gmail.com

**Keywords:** prostate cancer (PCa), RE-1 silencing transcription factor (REST), neuroendocrine PCa (NEPC), epithelial–mesenchymal transition (EMT)

## Abstract

The RE-1 silencing transcription factor (REST) is a repressor factor related to neuroendocrine prostate cancer (PCa) (NEPC), a poor prognostic stage mainly associated with castration-resistant PCa (CRPC). NEPC is associated with cell transdifferentiation and the epithelial–mesenchymal transition (EMT) in cells undergoing androgen deprivation therapy (ADT) and enzalutamide (ENZ). The effect of REST overexpression in the 22rv1 cell line (xenograft-derived prostate cancer) on EMT, migration, invasion, and the viability for ENZ was evaluated. EMT genes, Twist and Zeb1, and the androgen receptor (AR) were evaluated through an RT-qPCR and Western blot in nuclear and cytosolic fractions of REST-overexpressing 22rv1 cells (22rv1-REST). The migratory and invasive capacities of 22rv1-REST cells were evaluated via Transwell^®^ assays with and without Matrigel, respectively, and their viability for enzalutamide via MTT assays. The 22rv1-REST cells showed decreased nuclear levels of Twist, Zeb1, and AR, and a decreased migration and invasion and a lower viability for ENZ compared to the control. Results were expressed as the mean + SD of three independent experiments (Mann–Whitney U test, Kruskal–Wallis, Tukey test). REST behaves like a tumor suppressor, decreasing the aggressiveness of 22rv1 cells, probably through the repression of EMT and the neuroendocrine phenotype. Furthermore, REST could represent a response marker to ENZ in PCa patients.

## 1. Introduction

### 1.1. Prostate Cancer

Prostate cancer (PCa) is the second most common cancer in the world and represents the fifth leading cause of death in men. According to GLOBOCAN data, 1.5 million men were diagnosed with PCa in 2022 and approximately 397,430 deaths occurred due to this cancer [1].

Despite the screening strategies for the early detection of this cancer, such as measuring PSA levels in the blood and conducting a digital rectal examination [2,3], approximately 20% of patients present with advanced or metastatic disease at the time of diagnosis, in which the first-line treatment is androgen deprivation (ADT). These patients will progress to a phase with a worse prognosis, called castration-resistant prostate cancer (CRPC) [3,4], in a period between 18 and 24 months after starting ADT.

### 1.2. Castration-Resistant Prostate Cancer (CRPC)

CRPC is defined as PCa disease progression despite pharmacological or surgical castration. Under these conditions, the androgen axis continues to have a high influence on the function and growth of this cancer [5]. Understanding the transition from an androgen-sensitive cancer to a CRPC is a key challenge to generating therapies to control this disease.

Among the causal mechanisms for CRPC, the amplification of the androgen receptor (AR) gene, the generation of splicing variants, its promiscuity to non-androgenic signals and altered steroidogenesis, among others, have been described [4,5,6]. On the other hand, an interesting mechanism of androgen independence is the generation of neuroendocrine PCa (NEPC), which is characterized by the presentation of canonical markers, such as chromogranin A (CgA) and synaptophysin (SYP).

### 1.3. The Neuroendocrine PCa Subtype (NEPC)

NEPC has been proposed as the result of a transdifferentiation process occurring in the tumor cells that result in CRPC [7,8] and generally shows a histology of poorly differentiated small-cell neuroendocrine carcinoma or with a mixed neuroendocrine morphology. The current diagnosis of NEPC is performed using a metastatic tumor biopsy, and these patients may be considered for platinum-based chemotherapy. The average survival after NEPC diagnosis is seven months [9].

Although some NEPCs originate from neuroendocrine cells that normally reside in the prostatic epithelium, this event is extremely rare, representing approximately 2% of cases. Instead, in CRPC, the presence of NEPC is highly prevalent (40–100%) [10,11,12]. Furthermore, the NEPC phenotype becomes more recurrent in patients treated with first- and second-line AR pathway inhibitors. Therefore, when patients are treated with ADT, a certain negative pressure is applied so that the cancer progresses with the cells that can survive this therapy, which correspond to AR-negative cells and positive neuroendocrine markers [13,14]. Different pathologic subtypes of NEPC have been described in the context of CRPC [9].

### 1.4. The Androgen Receptor (AR)

AR is a ligand-dependent transcription factor that controls the expression of specific genes. The interaction of AR with testosterone or dihydrotestosterone through the ligand-binding domain initiates sexual development and differentiation [15]. AR splicing variants lack the ligand-binding domain and are constitutively active [16]. Among the most frequent splicing variants found in CRPC samples is AR-V7 [15]. In preclinical models, the expression of AR-V7 increases after castration conferring its primary and acquired resistance to abiraterone acetate and enzalutamide (ENZ) [17].

Studies on AR splicing variants in PCa have only been performed in samples from patients with PCa and CRPC, but not in NEPC, which has been characterized by the absence of AR, the presence of markers such as CgA and SYP, and the decrease in the RE-1 silencing transcription factor (REST).

### 1.5. The RE-1 Silencing Transcription Factor (REST)

REST is a factor that represses the neural phenotype in non-neural tissues [18]. The gene silencing mechanism used by REST consists of the recruitment of histone deacetylases, methyltransferases, and demethylases to their target genes through the interaction of their two repression domains with a specific 21 base pair sequence called response element 1 (RE-1) [19,20]. REST has been implicated in a number of pathologies, such as psychiatric disorders, neurodevelopment, and cancer [21]. In this latter context, REST has the role of an oncogene in neural tissues, and a tumor suppressor in non-neural tissues [22].

In the case of PCa, in 2012, REST was identified within genes with a low expression in samples with a neuroendocrine morphology. These findings resulted from the genomic characterization of a patient diagnosed with NEPC and were confirmed in their LNCaP cells, which were transfected with an siRNA against REST, causing a concomitant increase in neuroendocrine markers, such as SYP [23]. Moreover, REST has been related to various cellular processes involved in cancer progression, and in the case of PCa, it has been directly associated with the epithelial–mesenchymal transition (EMT) process.

### 1.6. The Epithelial–Mesenchymal Transition (EMT)

The EMT process plays a fundamental role in the normal development of the prostate gland in the embryonic period and tissue regeneration during wound closure, but it is also relevant in the tumor progression and development of phenotypes resistant to therapies [24,25]. The factors involved in the EMT process are involved in repressing molecules participating in a cell–cell interaction, such as E-cadherin, and stimulating the expression of those associated with the mesenchymal phenotype, such as vimentin and N-cadherin. Among the most relevant factors of the EMT is ZEB1, a factor related to the resistance to androgens and others chemotherapeutics in cellular models of PCa, and Snail, Slug, and Twist, as factors that repress the transcription of E-cadherin causing an increase in mesenchymal proteins [26,27].

To date, no research has evaluated the effects of REST on the expression profile of the four canonical EMT regulatory genes, or epithelial and mesenchymal markers.

This article shows the effects of REST overexpression in PCa cell lines on the activation of the EMT process and the malignant phenotype, through the evaluation of proliferation, migration, and invasion.

## 2. Results

### 2.1. The Expression of REST in Prostate Cancer Cell Lines

In order to choose the proper cell line that would be used to overexpress REST, the basal expression levels of four commercial PCa lines, LNCaP, 22rv1, PC3, and DU145, were studied. In agreement with our results, DU145 cells presented the highest levels of REST transcripts while 22rv1 and PC3 cells presented the lowest (Figure 1a,b).

On the other hand, the highest protein levels of REST were identified in the PC3 line and the lowest in 22rv1 (Figure 1c,d). Because REST is a factor found mainly in the nucleus, all protein studies were performed on nuclear and cytosolic fractions separately.

Additionally, we carried out an analysis of REST and EMT markers’ expression in PCa tumor samples using the GENT2 public database (Gene Expression database of Normal and Tumor tissues, http://gent2.appex.kr/gent2/, accessed on 22 February 2024) to further support our findings in the cell culture system. Interestingly, REST expression shows a significant decrease in higher Gleason grades, suggesting it has a role in NEPC risk. The results of this data are shown in Appendix A.

### 2.2. The Overexpression of REST Produces Modifications in the Expression Levels of EMT Regulators and Epithelial–Mesenchymal Markers

To determine if REST modifies the expression of the regulatory factors of the EMT, an overexpression of REST was carried out in the 22rv1 cell line through a lentiviral transduction vector which expressed the REST gene sequence, with a tag for hemagglutinin (HA) expression. The 22rv1 cells transduced with an empty lentiviral vector (NULL) were used as the control. The transduced 22rv1 cells showed a significant increase in REST expression at the transcript and protein levels. The latter was increased in both cellular fractions evaluated (Appendix A).

In our cellular model of REST overexpression, we evaluated the expression of Snail, Slug, Twist, ZEB1, CDH1, and VIM. An increase in Snail (the SNAI1 gene) transcript and protein levels (Figure 2a,b), and an increase in Slug (the SNAI2 gene) transcripts (Figure 2a) were observed. On the other hand, there was a significant decrease in the mRNA levels of ZEB1, CDH1, and VIM, with the latter being coincident with the protein level. Although the levels of Twist transcripts (the TWIST1 gene) show no significant differences (Figure 2a), there was a significant decrease in its nuclear protein level (Figure 2b,c).

### 2.3. The Effect of REST Overexpression on the Proliferation and Clonogenic Capacity of 22rv1 Cells

The activation of the EMT process produces a stimulation of the proliferation and self-renewal capacity of the cells, so these two parameters were evaluated in 22rv1 cells with an overexpression of REST. Figure 3a shows that the 22rv1 REST-HA cells present a significantly decreased proliferation compared to the control, while a similar phenomenon occurs in the evaluation of clonogenicity, showing a lower number of colonies in the cells with an overexpression of REST 14 days after the beginning of the experiment (Figure 3b).

### 2.4. The Overexpression of REST in 22rv1 Cells Decreases Migration and Invasion

Another of the main effects caused by the activation of the EMT program is the increase in the migratory and invasive capacities of cancer cells. In REST-overexpressing 22rv1 cells, a significant decrease in migration and invasion was observed (Figure 4a,b).

### 2.5. The Overexpression of REST in 22rv1 Cells Decreases Their Viability for Enzalutamide and Alters the Expression of AR and ARv7

One of the most widely used treatments of CRPC is with enzalutamide (ENZ). Therefore, the effects of REST overexpression on the viability of cells to this drug were evaluated. Figure 5a shows that 22rv1 REST-HA cells present lower viability for both DHT and ENZ, compared to NULL cells. Because ENZ is an AR inhibitor and ENZ resistance is associated with ARv7 expression, we determined the levels of these two markers. The expressions of the AR and ARv7 transcripts were not altered by REST overexpression (Figure 5b), but alterations at the protein level were observed. Interestingly, there was a significant decrease in nuclear AR, while an increase in the cytosolic fraction was observed. On the other hand, the ARv7 protein was decreased in both cell fractions. Additionally, the expression of KLK3 transcripts (gene that encodes the PSA protein) was decreased in cells with REST overexpression (Figure 5b).

### 2.6. Evaluation of EMT-Inducing Genes and Epithelial–Mesenchymal Markers

In the 22rv1 REST-HA lines, the SNAI1 transcript and protein were significantly increased. Instead, only the SNAI2 mRNA showed elevated levels. Twist showed a decreased expression, and ZEB1 showed decreased transcript and protein levels at the nuclear level, with no significant changes to the cytoplasmic fractions.

To achieve a better analysis of the findings, a study on the promoters of the four EMT regulators analyzed was carried out to identify if they have putative REST-binding sites. In addition, the presence of binding sites for EMT regulators in the REST gene was evaluated. For this study, the Eukaryotic Promotes Database available on the site http://epd.vital-it.ch was consulted (accessed on 24 April 2023).

A search for the promoter regions for REST, SNAI1, SNAI2, TWIST1, and ZEB1 in Homo sapiens was carried out. Once the genes were found, the analysis of the promoter sequences was performed using the Transcription Factor Motifs tool (JASPAR CORE 2018 vertebrates). Each of the predictions of the binding of a transcription factor was calculated with a *p* < 0.001 (Figure 6).

## 3. Discussion

The REST factor was evaluated in different PCa cell lines. The DU145 line presented the highest levels of the transcripts, and the PC3 line showed the highest levels of the protein in the nuclear fractions. These results are similar to those found in the literature regarding REST mRNA levels, coinciding with the DU145 line presenting the highest levels, but differing from Chang et al., regarding protein levels, who reported the LNCaP line to be presenting the highest levels [28]. The differences with Chang et al. may be due to the fact that their protein level evaluations were performed of in total extracts, and not in cell fractions.

An interesting observation is that PC3 showed the highest REST protein levels in the nuclear fraction. This cell line represents an advanced stage of the disease, being a commercial line not expressing AR and used as a model for the CRPC study. In the literature, a high protein expression of REST in the PC3 cell line has been reported [28], and also has been used as a model to analyze the effects of the induction of the neuroendocrine phenotype with dovitinib [29]. Exposure to this drug for a period of 3 weeks produced an increase in the neuroendocrine markers with a concomitant decrease in the expression of the REST transcript in LNCaP and PC3 cells. These results would support that decreasing REST levels in PC3 cells are sufficient to induce the transdifferentiation to a neuroendocrine phenotype [29]. According to searches carried out in The Human Protein Atlas database (available at https://www.proteinatlas.org/ [accessed on 1 June 2023]), PC3 cells show a high expression of the PRICKLE1 mRNA, encoding the RILP protein, which is responsible for REST’s translocation to the nucleus [30]. This could explain the nuclear concentration of REST found in PC3 [31]. Additionally, data from an analysis of REST and EMT markers’ expression in PCa tumor samples using the GENT2 public database support our findings in the cell culture system, showing a negative correlation between REST expression and Gleason grades, suggesting a role in NEPC risk.

Svensson et al. demonstrated that AR expression in PCa is related to high REST expression [32]. Therefore, cells showing high AR expressions are expected to show high REST, such as LNCaP [31], which is in agreement with our results. On the other hand, 22rv1 shows the lowest levels of REST among the androgen-dependent cells evaluated. This could be explained by the fact that these cells have the highest levels of SRMM4 and β-TRCP, inducing the alternative splicing of REST mRNA generating the non-functional variant REST4. In addition, β-TRCP could ubiquitinate the protein product and the consequent degradation via the proteasome (observations made in The Human Protein Atlas).

In relation to the different canonical EMT markers evaluated, the results of the Snail protein levels are interesting to analyze. In 2013, the regulation of miR-124a by REST was reported [33,34]. Also, the decrease of this miRNA increases the expression of Snail in glioblastoma multiforme [35]. The 22rv1 cells are the PCa cell lines presenting the highest levels of miR-124a (The Human Protein Atlas), suggesting that an increase in REST could cause a significant impact on the expression levels of this regulator. There is no information on this regulation in PCa, but this phenomenon cannot be ruled out since the behavior of REST and the repression machinery depends on the cellular context in which the evaluation is made [36]. Additionally, Snail has no putative REST-binding site, suggesting that REST exerts no direct action on Snail.

Snail changes are not related to the modification of vimentin expression. Instead, vimentin levels are more related to the changes observed in Twist and ZEB1. These findings suggest that REST could contribute to establishing incomplete EMT phenotypes, as has been evidenced in other studies. Some of these phenotypes have an expression of genes related to mesenchyme, but often conserving those associated with the epithelial phenotype, such as E-cadherin [37,38]. The increase in Twist1 is coincident with Chang et al.’s report, where a significant decrease in the protein levels of this transcription factor was observed in 22rv1 cells that overexpress REST [28]. Regarding ZEB1, to date, there is no report showing a direct relationship between this EMT inducer and REST. On the other hand, we show that overexpression of REST causes a decrease in the ZEB1 transcript without significant changes at the protein level. Interestingly, ZEB1 is expressed in both the nucleus and the cytoplasm. In 2022, Guo et al. reported that ZEB1 has different roles depending on its location, activating the EMT process when it is in the nucleus, but in the cytoplasm, it can bind to free actin monomers and RhoA, inhibiting migration, invasion, and proliferation [39], in accordance with our results.

On the other hand, the decrease in the level of ZEB1 transcripts can be explained by the action that REST exerts on the regulatory sites of gene expression, recruiting the repression machinery through its N- and C-terminal domains. According to our in silico analysis of promoters, this would be possible, since the ZEB1 gene has five putative binding sites for REST (Figure 6).

In addition to the changes described for the transcription factors regulating EMT, we evaluated the expression levels of vimentin, an intermediate filament characteristic of mesenchymal cells [40,41]. The promoter of the VIM gene has several binding sites for factors that activate and inhibit its transcription. Indeed, Sp1, upon binding to the VIM promoter, significantly increases its expression. On the contrary, preventing Sp1 from binding to the promoter causes decreased expression. This may be due to the binding of repressive factors to the promoter, remodeling the chromatin structure and making the Sp1 binding site less accessible. In studies carried out in neuroblastoma cell lines, it was observed that Sp1 increases the expression of synapsin I (SYN I), while REST prevents the binding of Sp1, decreasing SYN I [42]. A similar mechanism could be occurring in our model. On the other hand, the Sp1 promoter has several putative REST-binding sites, suggesting a REST-regulated Sp1 expression, indirectly impacting vimentin expression.

In our PCa models, the REST factor is capable of partially repressing the expression of the initiating factors of the EMT process by decreasing the expression of Twist and ZEB1 associated with the decrease in vimentin expression. The behavior observed at the Snail level is interesting, requiring new research to explain its increase in REST-HA lines. One hypothesis to consider is that in 22rv1 cells, REST could function as an activator of Snail under the mechanisms described above.

Interestingly, REST overexpression caused a decrease in AR at the nuclear level and an increase at the cytosolic level. This is in contrast to the observations made by Svensson, since this reduction is also accompanied by a low level of KLK3 transcripts [32]. The difference between the levels of REST in the different cellular fractions may be related to the translocation of AR to the nucleus. This process is related to the release of AR from its binding to HSP90, which is inhibited by the action of protein kinase A (PKA) [43]. It is possible that, by some mechanism, REST inhibits the translocation of AR to the nucleus. This is related to the results observed when REST-HA cells were incubated with DHT or ENZ (Figure 5). It is possible that the inhibition of AR translocation is highly effective, and that not even DHT incubation can increase viability levels, comparable to that of the control. In the case of the incubation with ENZ, the REST-HA lines showed significantly lower viability than the control cells, and could have an additive effect to the action that already occurs simply by overexpressing REST. These results are interesting because ENZ is the main second-line therapy for patients with CRPC and is applied prior to NEPC appearance. Therefore, a decreased viability in ENZ-treated 22rv1 cells could indicate an increase in therapy success for these patients [44,45].

The AR acts as a transcription factor for a series of genes, including some related to cell proliferation [46]. This transcription implies that AR is translocated to the nucleus through the action of PKA and is released from HSP90. Therefore, these facts are consistent with our results in 22rv1 cells with an overexpression of REST, since they present a significant decrease in its proliferation.

On the other hand, in the nuclear and cytosolic extracts of the 22rv1 lines with REST overexpression, there was a decrease in ARv7 levels. This functional AR splicing variant is one of the most frequent to appear in the context of androgen deprivation and its decrease could also account for the KLK3 messenger levels.

In 2014, Antonarakis et al. described that the expression of ARv7 in PCa would provide a resistance to ENZ and abiraterone acetate treatments [47]. These observations agree with our results, where cells with a high expression of REST presented low protein levels of ARv7 and a lower viability when incubated with ENZ.

Among the distinctive characteristics of cancer, proliferation is an essential process in its aggressive behavior [48]. We found that 22rv1 cells with REST overexpression showed lower proliferation compared to the control and a decrease in their colony formation capacity. In several studies on tumors of epithelial origin, the effect of REST on the proliferation process has been evaluated, suggesting some mechanism. One of these mechanisms includes the PI3K-Akt-signaling cascade. In one of these studies, it was observed that decreasing REST levels in colon cancer cells increased Akt phosphorylation [49], a relevant factor that leads cells to proliferate. Similar findings were described in small cell lung cancer cells [50]. Another mechanism proposed is the relationship of REST with LIN28A, a protein involved in the development and self-renewal processes of embryonic stem cells. In breast cancer cells, a decrease in REST expression causes an increase in LIN28A, which induces an increase in cell proliferation [51]. Taken together, all this evidence supports our results, raising a new research challenge to elucidate the REST mechanism for decreasing the proliferation and colony formation capacity in 22rv1 cells.

Additionally, among our results, we observed that in 22rv1 REST-HA cells, migration and invasion capacities decrease, associated with the decrease in the expression of Twist, ZEB1, and vimentin. Vimentin has been linked to cell motility. In studies where breast cancer cells were transfected with a vimentin overexpression vector, an increase in motility was observed, while silencing vimentin decreased it. Furthermore, a disorganization of cell–cell interactions was observed [40]. These results would contribute to explain our observations that cells with a REST overexpression decrease their migration. Vimentin is a characteristic intermediate filament of mesenchymal cells with a high capacity for migration and invasion, which, in the context of cancer, would result in favoring the metastasis process and worsening the patient prognosis [41]. The invasion of mesenchymal cells requires the degradation of the extracellular matrix through proteolytic enzymes such as matrix metalloproteinases (MMPs). In turn, the expression of MMPs is regulated by epithelial–mesenchymal transition factors [52], some of which decrease their expression in 22rv1 cells with a REST overexpression.

In breast cancer, a relationship between the expression of the Twist factor with MMP-2 and MMP-9 was described and evaluated via immunohistochemistry. It is proposed that Twist is a direct regulator of these two metalloproteinases [53]. This could contribute to explaining our results, since the decrease in Twist at the nuclear level would reduce its ability to transcribe these two metalloproteinases, thus reducing the invasion capacity.

Results from our laboratory have also shown that in PCa lines, ZEB1 is associated with the expression of MMP-2 and MMP-7 through the action of SPARC. In PC3 lines with SPARC silencing, a decrease in ZEB1 and MMP-2/-7 occurs, leading to a decrease in the migratory and invasive capacities of this cell line [54]. Because in our cells with an overexpression of REST, the migratory and invasive capacities as well as the expression of ZEB1 at the nuclear level are decreased, we could speculate that this may be accompanied by a decrease in both MMPs described. Additionally, in breast cancer, a regulatory role of REST on MMP24 expression, a gene that is increased in PCa, has been proposed. The silencing of REST causes an increase in the expression of MMP24, which has the consensus sequence for binding REST to its promoter, possibly regulating it [55]. It is possible that this also occurs in our cells, where an overexpression of REST decreases invasion.

Finally, our results suggest that REST has an inhibitory effect on the malignant phenotype, partially suppresses the EMT, and might be a potential treatment marker for ENZ.

## 4. Materials and Methods

### 4.1. Cell Cultures

The PCa lines were obtained from AddexBio Technologies (San Diego, CA, USA). LNCaP clone FGC and 22rv1 cells were cultured in a Roswell Park Memorial Institute (RPMI) 1640 medium (GIBCO, Life Technologies, Carlsbad, CA, USA), plus 2.5 g/L of D-glucose and 0.11 g/L of sodium pyruvate, according to the manufacturer’s recommendation. DU145 and PC3 cells were maintained in Dulbecco’s modified Eagle medium (DMEM) F12 (GIBCO, Life Technologies, Carlsbad, CA, USA). Both media were supplemented with 10% *v*/*v* fetal bovine serum (FBS; Mediatech, Manassas, VA, USA) and streptomycin-penicillin antibiotics (Corning Inc., Corning, NY, USA). The cultures were maintained at 37 °C, in a humidified environment with 5% CO_2_.

### 4.2. Lentiviral Transduction

For REST overexpression, cells were transduced with a lentivirus containing a REST sequence coupled to a hemagglutinin (HA) tag at its C-terminus (pLenti-U6-suCMV[hREST-HA]-Rsv[RFP-Puro]) or the empty vector as control (pLenti-suCMV[NULL control]-Rsv[RFP-Puro]). Lentivirals were obtained from GenTarget Inc. (San Diego, CA, USA), and the cells were infected following the manufacturer’s recommendations. The incubation was carried out with the lentivirals with a viral multiplicity factor of 3 (moi 3), for 72 h. After 72 h, the cells were washed with 1X PBS, and the transfected cells were incubated with puromycin at 2.5 µg/mL for 24 h. Cells in all conditions were subsequently subjected to selection by fluorescence-activated cell sorting (FACs). Cells with the overexpression are referred to as REST-HA, while controls are referred to as NULL.

### 4.3. RNA Extraction and RT-qPCR

Total RNA was obtained using TRIzol (Ambion, Life Technologies, Carlsbad, CA, USA). The TRIzol homogenate was centrifuged for 15 min at 10,000× *g* at 4 °C. Subsequently, the aqueous phase was recovered and mixed with concentrated isopropanol to precipitate the RNA. Subsequently, centrifugation was performed at 10,000× *g* for 10 min, obtaining a pellet at the bottom of the tube. The supernatant was carefully removed and washed once with 75% ethanol prepared with DEPC (diethyl pyrocarbonate) water. It was centrifuged again at 6000× *g* for 5 min, the new supernatant was removed and the pellets were left at room temperature until dry. The pellets were reconstituted with sterile nuclease-free water. The RNA was kept at −80 °C until use.

Determinations of the quantity and purity of the RNA were carried out in the Synergy HT equipment (Biotek Instruments, Winooski, VT, USA). In all experiments, samples that presented a ratio of approximately 2.0 were used. From the RNA, 2000 ng of cDNA were synthesized using the 5X All-In-One RT MasterMix kit (Vancouver, BC, Canada). An amount of 100 ng of cDNA was amplified using the Brilliant II SYBR Green qPCR Master Mix kit (Agilent Technologies, Santa Clara, CA, USA).

For amplification, a protocol (recommended by the manufacturer) was used with 10 min of initial denaturation at 95 °C, and 35 consecutive cycles of 15 s of denaturation at 95 °C, 15 s of annealing at the optimal temperature of the primers, and 15 s at 72 °C for elongation. To obtain the dissociation curve, an additional cycle was carried out at the end of the PCR reaction, where the samples were denatured for 10 s at 95 °C, hybridized at 70 °C for 1 s and a gradual increase in temperature was performed for 10 min until they reached 95 °C.

The constitutive gene GAPDH was used as a normalizer and the results were analyzed using the method ΔΔCt.

Primers in this study are presented in Appendix A.

### 4.4. Protein Extraction and Western Blot

For the identification of the proteins, cellular fractionation was carried out in order to have the nuclear and cytosolic proteins fraction separately. To do this, cells seeded in 60 mm plates at 80% confluency were washed twice with cold PBS and then lysed with a cytosolic fractionation buffer (CFB) (250 mM of sucrose, 20 mM of HEPES pH 7.4, 10 mM of KCl, 1.5 mM of MgCl2, 1 mM of EDTA, 1 mM of EGTA, 1 mM of DTT, plus protease inhibitors [cOmplete mini, EDTA-free, Roche, Basel, Switzerland]) with a cell scraper. After lysis with the CFB, the homogenate was transferred to a new 1.5 mL microcentrifuge tube and centrifugation was performed at 700× *g* for 5 min. The supernatant was collected in another 1.5 mL tube and saved for later processing. The resulting pellet was resuspended with 400 μL of CFB and centrifuged again at 700× *g* for 10 min. The supernatant was discarded and the pellet was resuspended in a nuclear fractionation buffer (NFB) (50 mM of Tris HCl pH 8.0, 150 mM of NaCl, 1% NP-40, 0.5% sodium deoxycholate, 0.1% *w*/*v* of SDS, plus protease inhibitors), thus obtaining the nuclear fraction. Before performing the quantification, the nuclear fraction was sonicated 2 times using 3 pulses at an amplitude of 30. The supernatant stored in previous steps was centrifuged at 10,000× *g* for 10 min, obtaining a new supernatant, which was collected in a new tube, thus obtaining the cytosolic fraction. The protein quantification of both fractions was performed using the Bradford method (Sigma Aldrich-Merck, Burlington, MA, USA) and the blots were made from 10 μg of protein from each of the fractions. Membranes were blocked with 5% *w*/*v* fat-free milk in a Tris buffer saline (TBS) with 0.2% Tween for 1.5 h and then incubated overnight with the primary antibodies diluted in blocking solution. After washing, the membranes were incubated with goat anti-mouse or rabbit anti-IgG (H + L) secondary antibodies, conjugated with peroxidase (HRP) and revealed with a chemiluminescence detection kit for HRP (EZ-ECL, Biological Industries, Cromwell, CT, USA). The results of the technique were developed and photographed with a Vilber Lourmat capture system (Fusion FX5-XT 826.WL/Superbright, Vilber). Tubulin was used as the loading control of the cytosolic fraction and histone H3 was that of the nuclear fraction. Semiquantification analyses of the generated bands were performed with the ImageJ 1.51w program (NIH, Bethesda, MD, USA).

The antibodies used in this work were anti-REST (1:1000, abcam, ab21635), anti-Snail (1:1000, Cell Signaling Tech, 3879), anti-Twist (1:1000, Sigma-Aldrich, T6451), anti-ZEB1 (1:1000, eBiosciencie, Themo Fisher, 14974182), anti-E-cadherin (BD Biosciences, 610181), anti-vimentin (1:1000, abcam, ab8978), anti-histone H3 (1:10,000, abcam, ab1791), anti-HA (1:1000, abcam,, H6908), anti-RFP (1:10,000, ChromoTek, 6G6), anti-AR (1:1000, abcam, ab9474), anti-ARv7 (1:1000, abcam, ab198394), anti-tubulin (1:10,000, Sigma Aldrich, T0198), goat anti-mouse (1:10,000, Jackson Immunoresearch Inc.), and goat anti-rabbit (1:10,000, Jackson Immunoresearch Inc., West Grove, PA, USA).

### 4.5. Clonogenicity Assays

To evaluate the characteristics associated with cellular malignancy, the colony formation assay was used. A number of 2000 cells corresponding to the 22rv1 NULL and 22rv1 REST-HA lines were seeded in 6-well plates. Cultures were maintained for 15 days, with a medium change every 2 days. Each experiment was performed in triplicate for each of the cellular conditions evaluated. Once the colonies were obtained, the cells were fixed with a 4% *w*/*v* paraformaldehyde buffered at pH 7.1 for 30 min at room temperature. The fixation was washed with 1X PBS. Colony staining was performed in a 0.2% *w*/*v* aqueous crystal violet for 10 min. Once stained, the plates were allowed to dry at room temperature.

### 4.6. Transmigration and Invasion Assays

The 22rv1 cells transfected with REST-HA were seeded in 24-well Transwell^®^ chambers with a pore size of 8 µm (Sigma-Aldrich, St. Louis, MO, USA, Cat#CLS3422). A number of 5000 cells were seeded per well for the transmigration assay and 7500 for the invasion assay. For this last test, Matrigel (Sigma-Aldrich, San Louis, MO, USA, Cat#E1270) was added, which was left on the plates overnight at 4 °C and subsequently the plates were allowed to gel for 30 min at 37 °C before sowing. The invasion of the cells in the Matrigel was carried out by adding 10% fetal bovine serum to the lower chamber. Migration and invasion were evaluated after 24 h in a culture incubator at 37 °C and 5% CO_2_. The cells that did not migrate or invade were removed, and those adhered to the Transwell^®^ membrane were fixed and stained at the same time with a 0.2% *w*/*v* crystal violet solution, dissolved in 10% *v*/*v* methanol. Migrating and invasive cells were counted and expressed in the graphs as the average of the count in each case. Each of these experiments was carried out three times, independently.

### 4.7. Evaluation of the Viability of Enzalutamide

Cell viability was assessed using 3-(4,5-dimethylthiazol-2-yl)-2,5-diphenyltetrazolium bromide (MTT). To do this, the NULL and REST-HA cells were synchronized with an RPMI 1640 medium without phenol red and without serum for 24 h. Subsequently, the cells were trypsinized with trypsin prepared in 1X PBS, and then counted in an automatic counter (TC Automated Cell Counter, Bio-Rad Laboratories, Hercules, CA, USA). A number of 5000 cells were seeded in a 96-well plate to evaluate their proliferation and in other plates to evaluate their viability for enzalutamide. At 48 h after seeding, cells were incubated for 30 min with enzalutamide (ENZ) at a concentration of 10 μM (ENZ group), or with 1 nM of dihydrotestosterone (DHT). Both enzalutamide and DHT were prepared in dimethyl sulfoxide (DMSO). For the control group, DMSO was added for a final concentration of 1% *v*/*v*. After 24, 48, and 72 h, the cells were washed with 1X PBS, and 100 mL of an MTT working solution (1 mL of MTT [5 mg/mL] in 9 mL of serum-free RPMI 1640 medium) was added and incubated for 3 h at 37 °C. Subsequently, the supernatant was discarded, the formazan crystals were resuspended in a 20% *v*/*v* DMSO solution in isopropanol, and the absorbance was measured at 570 nm in a BioTek Synergy HT plate reader (BioTek Instruments, Winooski, VT, USA).

### 4.8. Statistical Analysis

Data analysis was performed with the GraphPad Prism 6.0 program (GraphPad Software, La Jolla, CA, USA). In all experiments, data were represented as the mean ± the standard deviation from at least three independent experiments. The non-parametric Mann–Whitney test was used to determine the existence of significant differences between the measurements of the control and the experimental groups. For the MTT analyses, the two-way ANOVA test with Tukey’s multiple comparisons test was applied. In all cases, a *p* < 0.05 was considered statistically significant. The p value, the number of experiments, and the statistical test used in each particular case are detailed in the legends of the figures.

## 5. Conclusions

Our results support the fact that, in our model, REST behaves like a tumor suppressor, decreasing the aggressive behavior of 22rv1 cells, probably through the repression of the EMT process and the neuroendocrine phenotype. Furthermore, REST could represent a marker of response to ENZ in patients with PCa.

## 6. Limitations of the Study

Conclusions of this work should be taken with care and caution because they are based in cell culture experiments using commercial cell lines. It would be valuable to complement the results with an in vivo model. However, these kinds of models are rather complicated and need careful standardization to validate reproducibility. We are currently developing a transgenic mice model for human CRPC in which we will study the role of REST. On the other hand, it would also be interesting to carry out a complementary immunohistochemical study comparing AR and REST in the PCa cancer tissue of non-metastatic patients with a CRPC and NEPC (before and after treatment with AR inhibitors). Unfortunately, such a study in human biopsies is very difficult to carry out in terms of time. Currently, there is a significant lack of samples from patients during treatment with ADT and diagnosed with NEPC after anti-AR therapies. We are currently working on recruiting biopsies from those groups of patients at different stages of therapy.

## Figures and Tables

**Figure 1 ijms-25-03332-f001:**
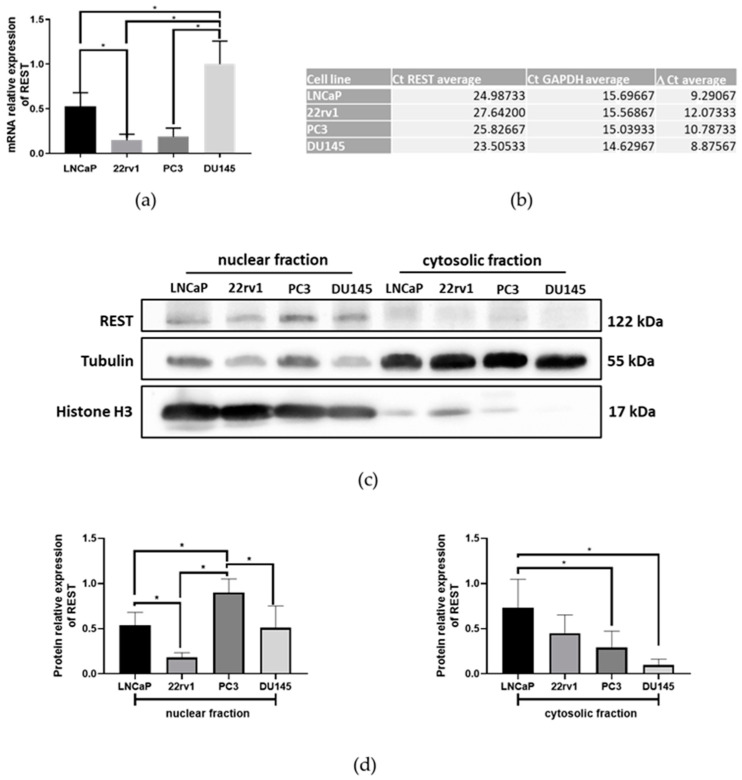
Expression of REST transcripts and protein levels in prostate cancer cell lines. (**a**) REST mRNA levels in LNCaP, 22rv1, PC3, and DU145 cells. REST levels were normalized to GAPDH expression; (**b**) Table of values of the average amplification cycle (Ct) of REST and GAPDH, and the average of the difference of both Cts for each of the cell lines evaluated; (**c**) Protein expression of REST in the nuclear and cytosolic fractions of the cell lines. The loading control for the nuclear fraction is histone H3 and for the cytosolic fraction is tubulin (**d**). On the right is the semiquantification graph of the REST in nuclear fractions and on the left is the graph of protein levels of the cytosolic fraction. In the Mann–Whitney U test, *p* values < 0.05 are considered significant (*), N = 4.

**Figure 2 ijms-25-03332-f002:**
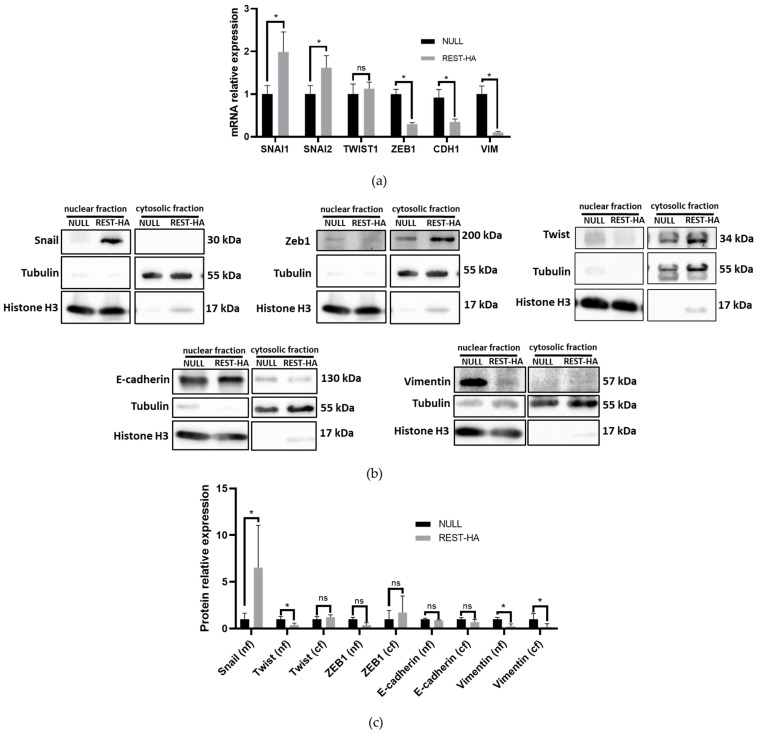
The effect of REST overexpression on the expression of EMT transcription factors, and of E-cadherin and vimentin. (**a**) Transcript levels of SNAI1, SNAI2, TWIST1, ZEB1, CDH1, and VIM in 22rv1 REST-HA and NULL cells; (**b**) Protein expression of Snail, ZEB1, Twist, E-Cadherin, and vimentin in nuclear and cytosolic fractions of 22rv1 RES-HA and NULL cells. (**c**) Graphs of protein levels of EMT, E-cadherin, and vimentin transcription factors. In the Mann–Whitney U test, *p* values < 0.05 (*) are considered significant, ns: not significant, N = 4.

**Figure 3 ijms-25-03332-f003:**
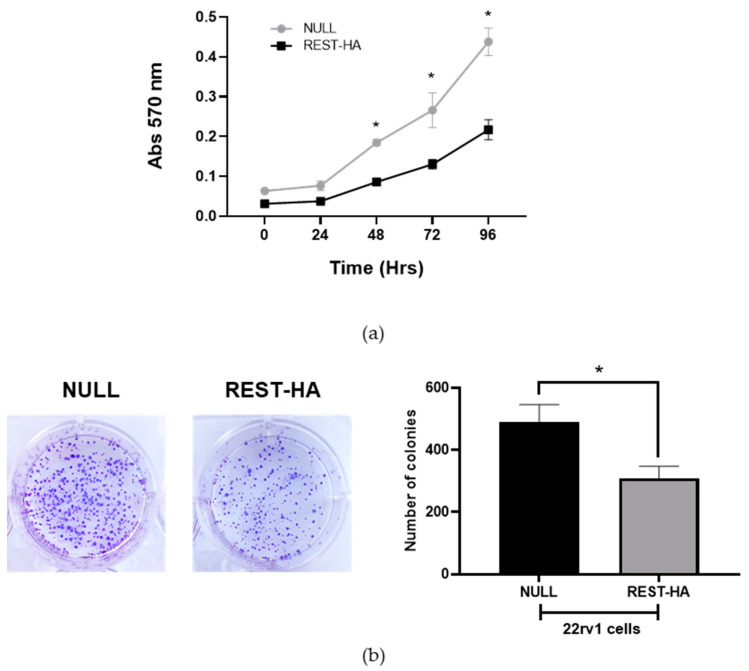
The proliferation and colony-forming capacity of cells with REST overexpression. (**a**) The proliferation evaluation of the 22rv1 REST-HA and NULL cells every 24 h after seeding. The two-way ANOVA test with Tukey’s multiple comparisons, N = 3; (**b**) Crystal violet-stained colonies obtained from the growth of 22rv1 REST-HA and NULL cells (**left**), and the quantification of the number of colonies of each cell line (**right**). In the Mann–Whitney U test, *p* values < 0.05 are considered significant (*), N = 4.

**Figure 4 ijms-25-03332-f004:**
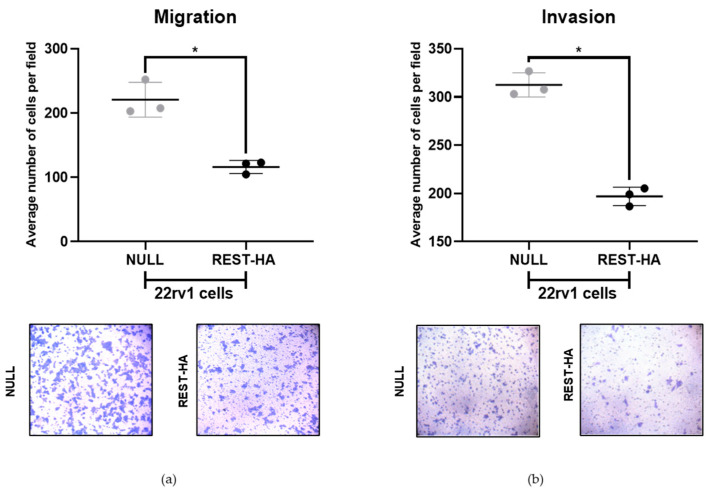
The effects of REST overexpression on the migration and invasion of 22rv1 cells. (**a**) The migration of 22rv1 cells assessed via Transwell assays. Migrating cells were stained with crystal violet (**bottom**) and quantified (**top**); (**b**) The invasion of 22rv1 cells evaluated via Transwell assays, with wells covered with Matrigel. Migrating cells were stained with crystal violet (**bottom**) and quantified (**top**). In the Mann–Whitney U test, *p* values < 0.05 are considered significant (*), N = 3. Microscope magnification: 400×.

**Figure 5 ijms-25-03332-f005:**
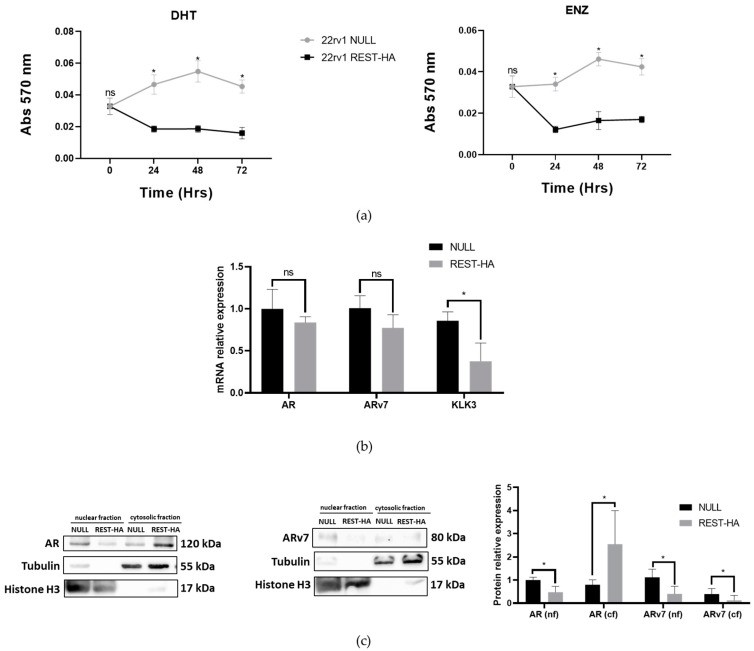
The evaluation of the response to DHT and enzalutamide and the expression of AR, ARv7, and PSA in cells with REST overexpression. (**a**) The assessment of the viability of 22rv1 REST-HA cells exposed to 1 nM of DHT (**left**) or 10 μM of ENZ for 72 h. The evaluations were carried out using the reduction capacity of the MTT. Two-way ANOVA test with Tukey’s multiple comparisons, N = 3; (**b**) Transcript levels of AR, ARv7, and KLK3 (PSA) in 22rv1 REST-HA cells. In the Mann–Whitney U test, *p* values < 0.05 are considered significant, N = 4. (**c**) Protein levels of AR and ARv7 in nuclear and cytosolic fractions of the 22rv1 REST-HA and NULL cells (**left**), and quantifications of the protein levels (**right**). In the Mann–Whitney U test, *p* values < 0.05 are considered significant (*), ns: not significant. N = 3.

**Figure 6 ijms-25-03332-f006:**
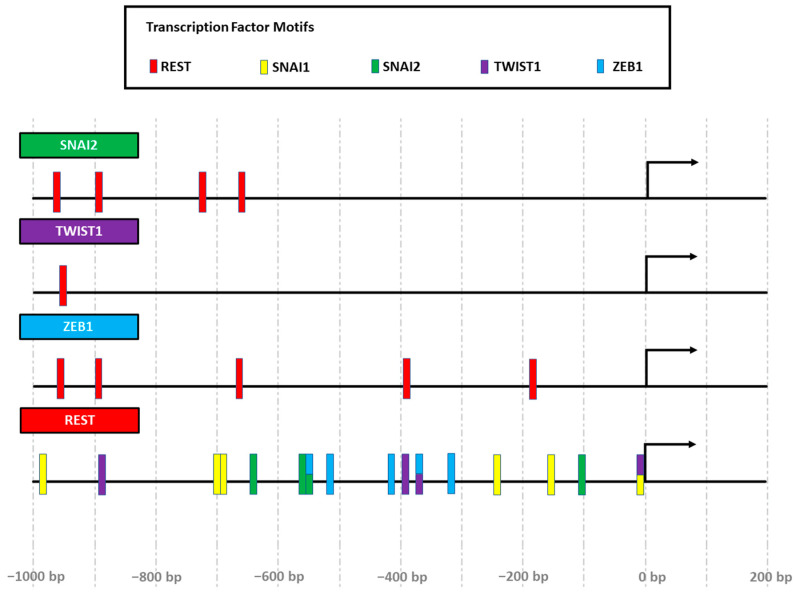
EMT regulatory genes contain REST binding elements, and REST presents binding elements for EMT regulators. The promoter regions of the genes SNAI1, SNAI2, TWIST1, ZEB1, and REST are represented from 1000 bp upstream to 200 bp downstream of the transcription start site (represented by arrows). Putative binding sites are represented in red rectangles for REST, green for SNAI2 (Slug), purple for TWIST1, and blue for ZEB1. Rectangles with different colors indicate the possible binding of one or more factors to the same site in the gene promoters. Binding sites are approximated and were obtained from the Eukaryotic Promotes Database (http://epd.vital-it.ch) using the JASPAR tool (accessed on 24 April 2023). Each of the binding sites is calculated with a *p* < 0.001.

## Data Availability

All supporting data of this manuscript are available upon request to the corresponding authors.

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
