# Peer review of "Overexpression of REST Represses the Epithelial–Mesenchymal Transition Process and Decreases the Aggressiveness of Prostate Cancer Cells"

_ijms, 2024, doi:10.3390/ijms25063332_

Round 1

Reviewer 1 Report

Comments and Suggestions for Authors

Dear Editor,

the manuscript of Indo S. et al. entitled Overexpression of REST Represses the Epithelial-Mesenchymal Transition Process and Decreases the Aggressiveness of Prostate Cancer Cells concerns a very complex issue about prostate cancer with worse prognoses. Indeed, this study focuses on Castration-Resistant Prostate Cancer (CRPC) and its Neuroendocrine Prostate Cancer (NEPC) subtype. Specifically, authors aim to evaluate the effects of different level of REST expression in different Prostate Cancer (PCa) cell lines on the Epithelial-Mesenchymal Transition (EMT), trying to study if REST may be considered as a new prognostic biomarker.

Even though the paper is written in a perfect English style and has a structure, it shows some strong weaknesses points that need to be revised. Since the topic is attractive and of current scientific interest, it is suggested to the authors to improve the present form of the manuscript as follow, in order to proceed with its publication on International Journal of Molecular Science (IJMS).

• In general, the study appears very complex but not very comprehensible, without a common thread that helps the reader to clearly understand the purpose of the work, considering the complete and complex knowledge about prostate cancer. For example, authors may organize the introduction of the manuscript with different sections: clinical features of CRPC e NEPC, their epidemiologic data and incidence, the state of the art about the knowledge on CRPC e NEPC (already partially illustrated), thus, concluding with the aim of the study. The introduction is limited: the authors did not even refer to the basic distinction between castration-resistant prostate adenocarcinoma (CRPC-adeno) and neuroendocrine prostate cancer (NEPC). It is recommended to the authors to improve in a significant way the introduction, trying to explain everything that is mentioned in the manuscript.

• Also, in the last sentence of the abstract, the authors refer to REST as a possible response marker to ENZ in PCa patients. This sentence is reported again in conclusion without any deepening in the discussion section and no link to the results obtained regarding its possible predictive value.

• It would be very interesting and attractive if authors would highlight why REST may be so important: why REST can make the difference in differential diagnosis or prognostic or predictive evaluations?

• The quality of the NULL and REST-HA Figures 3b, 4a and 4b needs to be improved.

• In order to make the study more comprehensive the authors may improve the study by adding an immunohistochemical section testing AR and REST, comparing prostatic cancer tissue of a non-metastatic patient with a CRPC and NEPC (before and after treatment AR inhibitors). Also, the main factors involved in EMT process can be tested by immunohistochemistry. In this case you should add a new section in Materials and Methods about tissues enrolment and indicate also the protocol number for the ethics committee approval.

Author Response

Reviewer 1:

Comment: The manuscript of Indo S. et al. entitled Overexpression of REST Represses the Epithelial-Mesenchymal Transition Process and Decreases the Aggressiveness of Prostate Cancer Cells concerns a very complex issue about prostate cancer with worse prognoses. Indeed, this study focuses on Castration-Resistant Prostate Cancer (CRPC) and its Neuroendocrine Prostate Cancer (NEPC) subtype. Specifically, authors aim to evaluate the effects of different level of REST expression in different Prostate Cancer (PCa) cell lines on the Epithelial-Mesenchymal Transition (EMT), trying to study if REST may be considered as a new prognostic biomarker. Even though the paper is written in a perfect English style and has a structure, it shows some strong weaknesses points that need to be revised. Since the topic is attractive and of current scientific interest, it is suggested to the authors to improve the present form of the manuscript as follow, in order to proceed with its publication on International Journal of Molecular Science (IJMS).

Answer: Thanks to the reviewer for the general appreciation of our work.

Comment: In general, the study appears very complex but not very comprehensible, without a common thread that helps the reader to clearly understand the purpose of the work, considering the complete and complex knowledge about prostate cancer. For example, authors may organize the introduction of the manuscript with different sections: clinical features of CRPC e NEPC, their epidemiologic data and incidence, the state of the art about the knowledge on CRPC e NEPC (already partially illustrated), thus, concluding with the aim of the study. The introduction is limited: the authors did not even refer to the basic distinction between castration-resistant prostate adenocarcinoma (CRPC-adeno) and neuroendocrine prostate cancer (NEPC). It is recommended to the authors to improve in a significant way the introduction, trying to explain everything that is mentioned in the manuscript.

Answer: We thanks and appreciate the reviewer comments. We have organized the introduction in different sections, including more details about CPCR and especially NEPC. We have included a new reference with the state-of-art of NEPC, including pathological description, for further information. (Yamada and Beltran Curr Oncol Rep. 2021; 23: 15). Modifications in the revised manuscript are highlighted in yellow.

Comment: Also, in the last sentence of the abstract, the authors refer to REST as a possible response marker to ENZ in PCa patients. This sentence is reported again in conclusion without any deepening in the discussion section and no link to the results obtained regarding its possible predictive value.

Answer: We thank the reviewer comment and have further explained our suggestion in the revised discussion, adding additional references to support it. Changes are highlighted in yellow.

Comment: It would be very interesting and attractive if authors would highlight why REST may be so important: why REST can make the difference in differential diagnosis or prognostic or predictive evaluations?

Answer: We agree with the referee in terms of the attractive idea that REST may make the difference in prognosis. However, we have been very careful and cautious in our conclusion because are based only in cell culture experiments. We are currently developing a transgenic mice model for human CRPC in which will study the role of REST.

Comment: The quality of the NULL and REST-HA Figures 3b, 4a and 4b needs to be improved.

Answer: We agree with the reviewer and have improved the indicated figures.

Comment: In order to make the study more comprehensive the authors may improve the study by adding an immunohistochemical section testing AR and REST, comparing prostatic cancer tissue of a non-metastatic patient with a CRPC and NEPC (before and after treatment AR inhibitors). Also, the main factors involved in EMT process can be tested by immunohistochemistry. In this case you should add a new section in Materials and Methods about tissues enrolment and indicate also the protocol number for the ethics committee approval.

Answer: We really appreciate the referee recommendation and agree with that study would add valuable complementary information to our work. Unfortunately, such study in human biopsies is very difficult to carry out in terms of time. As indicated by Wang et al (1), there is currently a significant lack in samples from patients during treatment with ADT and when NEPC is diagnosed after anti-AR therapies. We are currently working on recruiting biopsies from patients at different stages of therapy, but having a critical mass of samples of non-metastatic CRPC and NEPC before and after administration of various therapies takes a long time. We expected to finish recruiting this material by the end of this year or by the beginning of the next.

1.Wang Y, Wang Y, Ci X, Choi SYC, Crea F, Lin D, Wang Y. Molecular events in neuroendocrine prostate cancer development. Nat Rev Urol. 2021;18:581-596. doi: 10.1038/s41585-021-00490-0

Reviewer 2 Report

Comments and Suggestions for Authors

Dear Authors,

ijms-2879067

Overexpression of REST represses the epithelial-mesenchymal transition process and decreases the aggressiveness of prostate cancer cells by Indo et al summarizes the role of REST in NEPC. It is an interesting and comprehensive story. However, the study focuses mainly on in vitro experiments. The authors could show overexpression REST in NEPC cells that inhibits proliferation of NEPC in vivo will improve the manuscript quality.

Comments on the Quality of English Language

Minor English needed

Author Response

Reviewer 2:

Comment: Overexpression of REST represses the epithelial-mesenchymal transition process and decreases the aggressiveness of prostate cancer cells by Indo et al summarizes the role of REST in NEPC. It is an interesting and comprehensive story. However, the study focuses mainly on in vitro experiments. The authors could show overexpression REST in NEPC cells that inhibits proliferation of NEPC in vivo will improve the manuscript quality.

Answer: We appreciate the reviewer comment and agree about the importance of in vivo studies. Indeed, we are currently working in developing a transgenic mice model of CRPC to further studies on the role of REST. But these kinds of models are rather complicated and need careful standardization to validate reproducibility. We expected to have results from our in vivo model in the near future.

Comments on the Quality of English Language Minor English needed

Answer: Thanks for the observation. We have revised the English and corrected spelling found.

Reviewer 3 Report

Comments and Suggestions for Authors

the authors present a nice set of data demonstrating the role of REST in neuroendocrine prostate cancer. The experiments clearly make the point, they are clear cut and there is no questioning the results and conclusions. There are two minor points that would improve the quality of the manuscript. The first is that the authors in the introduction mention the GLOBOCAN data 2020. From February 8th 2024 the GLOBOCAN data 2022 is available. It would be nice if they could make the substitution (there are no major changes) and mention the appropriate article for these data (instead of the 2020).

The second point that would help to improve the manuscript would be the analysis of the REST, Snail, Twist, Zeb1, E-cadherin and vimentin in tumor samples from public databases such as GENT2 or others, aiming to corroborate the data they find using cell lines in actual human tumor samples.

Author Response

Reviewer 3

Comment: The authors present a nice set of data demonstrating the role of REST in neuroendocrine prostate cancer. The experiments clearly make the point, they are clear cut and there is no questioning the results and conclusions. There are two minor points that would improve the quality of the manuscript. The first is that the authors in the introduction mention the GLOBOCAN data 2020. From February 8th 2024 the GLOBOCAN data 2022 is available. It would be nice if they could make the substitution (there are no major changes) and mention the appropriate article for these data (instead of the 2020).

Answer: We appreciate the reviewer comment. We have updated de information of GLOBOCAN to 2024 and change de reference properly.

Comment: The second point that would help to improve the manuscript would be the analysis of the REST, Snail, Twist, Zeb1, E-cadherin and vimentin in tumor samples from public databases such as GENT2 or others, aiming to corroborate the data they find using cell lines in actual human tumor samples.

Answer: We thank the reviewer suggestion. We have carried out the requested analysis using GENT2 database focusing in REST and the indicated EMT markers in PCa tumor samples. Level of marker expression and tumor progression is showed. The Results section has been modified accordingly and results showed in new Supplementary Figure 2.

Round 2

Reviewer 1 Report

Comments and Suggestions for Authors

Dear Editor,

the manuscript of Indo S. et al. entitled Overexpression of REST Represses the Epithelial-Mesenchymal Transition Process and Decreases the Aggressiveness of Prostate Cancer Cells has been improved by the authors.

In order to proceed with its publication on International Journal of Molecular Science (IJMS), it is suggested to consider the following two points. The first is a crucial recommendation, whereas the second one is a suggestion:

• Since authors show the results on REST starting only from in-vitro experiments and stated that they were currently unable to confirm the results by other analyses, it is strictly recommended to add a new section on study limitations; here, authors can report all the weaknesses which may influence the conclusions of this research, including those mentioned by me and also by the Referee #2.

• Check the length of sentences. For example, the sentence added in the discussion section: “These results are interesting because 330 ENZ is the second-line therapy most applied to patients with CRPC and one of the 331 therapies administrated prior to the appearance of NEPC, therefore, a decreased viability 332 in 22rv1 cells treated with ENZ could indicate an increased success rate of this therapy in 333 these patients (44, 45).” is too long. It can be reported as follows: “These results are interesting because 330 ENZ is the second-line therapy most applied to patients with CRPC and one of the 331 therapies administrated prior to the appearance of NEPC; therefore, a decreased viability 332 in 22rv1 cells treated with ENZ could indicate an increased success rate of this therapy in 333 these patients (44, 45).”.

Author Response

Comment: Since authors show the results on REST starting only from in-vitro experiments and stated that they were currently unable to confirm the results by other analyses, it is strictly recommended to add a new section on study limitations; here, authors can report all the weaknesses which may influence the conclusions of this research, including those mentioned by me and also by the Referee #2.

Answer: Thanks to the reviewer for the recommendation. We have added, after conclusion section, a new section entitled “Limitations of the Study” in which we explain the care and caution with the conclusions should be taken.

Comment: Check the length of sentences. For example, the sentence added in the discussion section: “These results are interesting because 330 ENZ is the second-line therapy most applied to patients with CRPC and one of the 331 therapies administrated prior to the appearance of NEPC, therefore, a decreased viability 332 in 22rv1 cells treated with ENZ could indicate an increased success rate of this therapy in 333 these patients (44, 45).” is too long. It can be reported as follows: “These results are interesting because 330 ENZ is the second-line therapy most applied to patients with CRPC and one of the 331 therapies administrated prior to the appearance of NEPC; therefore, a decreased viability 332 in 22rv1 cells treated with ENZ could indicate an increased success rate of this therapy in 333 these patients (44, 45).”.

Answer: We agree with the reviewer and have shortened the sentence from 50 to 38 words including a period in the middle.